# Predicting Tree-Related Microhabitats by Multisensor Close-Range Remote Sensing Structural Parameters for the Selection of Retention Elements

**Julian Frey** [1,2,*] **, Thomas Asbeck** [3] **and Jürgen Bauhus** [3]

1 Chair of Remote Sensing and Landscape Information Systems, University of Freiburg,
D-79106 Freiburg, Germany
2 Chair of Forest Growth, University of Freiburg, D-79106 Freiburg, Germany
3 Chair of Silviculture, University of Freiburg, D-79106 Freiburg, Germany;
thomas.asbeck@waldbau.uni-freiburg.de (T.A.); juergen.bauhus@waldbau.uni-freiburg.de (J.B.)
* Correspondence: Julian.frey@iww.uni-freiburg.de; Tel.: +49-761-203-3733

**Abstract:** The retention of structural elements such as habitat trees in forests managed for timber production is essential for fulfilling the objectives of biodiversity conservation. This paper seeks to predict tree-related microhabitats (TreMs) by close-range remote sensing parameters. TreMs, such as cavities or crown deadwood, are an established tool to quantify the suitability of habitat trees for biodiversity conservation. The aim to predict TreMs based on remote sensing (RS) parameters is supposed to assist a more objective and efficient selection of retention elements. The RS parameters were collected by the use of terrestrial laser scanning as well as unmanned aerial vehicles structure from motion point cloud generation to provide a 3D distribution of plant tissue. Data was recorded on 135 1-ha plots in Germany. Statistical models were used to test the influence of 28 RS predictors, which described TreM richness ($R^2$: 0.31) and abundance ($R^2$: 0.31) in moderate precision and described a deviance of 44% for the abundance and 38% for richness of TreMs. Our results indicate that multiple RS techniques can achieve moderate predictions of TreM occurrence. This method allows a more efficient and objective selection of retention elements such as habitat trees that are keystone features for biodiversity conservation, even if it cannot be considered a full replacement of TreM inventories due to the moderate statistical relationship at this stage.

**Keywords:** forest biodiversity; tree related microhabitats; terrestrial laser scanning; UAV; structure from motion; forest structure

## 1. Introduction

Forests are enormously important for the conservation of biodiversity and the provisioning of habitats within forests is closely related to their structural richness or complexity. Forest structure, therefore, is an important driver for biodiversity among other forest ecosystem services [1–3]. Consequently, forest biodiversity conservation has shifted from a focus on single-species protection towards understanding and conserving multi-taxon as well as structural indicators of forest biodiversity [1,4–7] and forest taxa on different scales including fine-scale structures at the tree-level [1]. How to quantify forest structure has therefore become an important challenge for predicting habitat quality or monitoring forest biodiversity [8–10], yet, the understanding of forest structure and structural complexity differs to some extent between individual scientific disciplines. Generally, forest sciences focus on forestry variables such as diameter at breast height (DBH), tree species, basal area, canopy cover and age structure [8–11], or the number of standing dead trees indicating horizontal heterogeneity [12]. The remote sensing (RS) discipline mainly describes forest structure by summarizing variables that can

be determined from sensor data. These include maximum height, quantiles of height from surface models or point clouds, point densities, or structural complexity indices, tree counts, biomass estimates and many more [3,13–26]. If the aim is to provide a broader perspective of the forest structure, metrics such as the Stand Structural Complexity Index (SSCI) [4] are for instance applicable.

In forest management, there is a consensus that tree-related microhabitats (TreMs) are decent descriptors of habitat provision and hence are important indicators of forest biodiversity [1,27]. The current definition for a TreM is "a distinct, well delineated structure occurring on living or standing dead trees, that constitutes a particular and essential substrate or life site for species or species communities during at least a part of their life cycle to develop, feed, shelter or breed" [27]. The main taxonomic groups that have been addressed in the established typology of TreMs include invertebrates such as insects, arachnids and gastropods as well as vertebrates such as birds, rodents, bats and carnivores [27]. It is a matter of ongoing research to evaluate which taxa profit directly or indirectly from the provision of TreMs, so far there is evidence that especially bats, saproxylic beetles and birds are related to the occurrence of TreMs [1,28–31]. Habitat trees are considered large, old trees that offer a high number of TreMs compared to the "average" tree in a forest managed for timber production [32]. The selection of habitat trees based on TreMs has lately been implemented in various regions of Central Europe managed under continuous-cover forestry and therefore deserves increased attention [29,33–35].

TreM field inventories are commonly used to assess the quality and quantity of habitat trees [34,36] as remote sensing is not yet able to detect the full set of these relatively small structural attributes directly at the tree-level [37,38]. Rehush et al. [38] detected TreMs only on the stem section and just for beech (*Fagus sylvatica* L.) trees with advanced machine learning techniques and were able to detect those TreMs with an accuracy of up to 83%. However, close-range remote sensing techniques are able to describe small-scale structural complexity of forests in increasing detail [4,39–42]. TreM inventories and RS techniques are often used to answer similar research questions, for instance the quantification of old-growth attributes in forests [39,43] and the selection of habitat trees [32,44]. If remotely-sensed data can predict the range of forest structures that offer a high abundance and richness of TreMs, this would constitute a major step towards a more efficient and objective selection of habitat trees as retention elements for biodiversity conservation. In addition it could eliminate or reduce the need for labor-intensive field surveys of TreMs. Remote sensing data collection is more time- and cost-efficient and has less observer bias than traditional TreM field surveys [45].

To guide the selection of habitat trees, the present study analyzed the abundance and richness of TreMs measured in the field in relation to fine-scale structural variables that can be detected from close-range remote sensing. Our research question is to predict TreM abundance and richness, provided by ground-based assessments of TreMs, by parameters derived from close-range RS metrics.

## 2. Materials and Methods

### 2.1. Research Site

The study area is located in South-West Germany in the southern Black Forest mountain range in the state of Baden-Württemberg (Figure 1). The Black Forest rises from the Rhine valley up to ca. 1500 m a.s.l. at the highest peaks. The research project "Conservation of Forest Biodiversity in Multiple-Use Landscapes of Central Europe" (ConFoBi) established a network of 1-hectare plots in existing state-owned forests (Figure 1) [30]. Plots were selected following a procedure to ensure the independence of single plots by including a minimum distance of 750 m between the plots and to ensure gradients of forest connectivity and structure. The first gradient was the proportion of forest in the 25 km$^2$ surrounding of the plots and the second gradient was the number of standing dead trees per plot (see [30] for details). Forests in this area are dominated by Norway spruce (*Picea abies* L.), European beech (*Fagus sylvatica* L.) and silver fir (*Abies alba* Mill.). A full list of plots, their altitude and their tree species with the respective basal areas can be found in Table A2. Management of these forests follows a "close-to-nature" paradigm characterized by single tree and group selection harvests, natural

regeneration, promotion of mixed and uneven-aged stands, and retention of habitat trees [46]. The plots were selected so that water bodies, roads, power lines etc. were excluded but smaller man-made objects like raised hides for hunting, skid tracks, hiking paths etc. could be included. The plots cover a range of altitudes between 434 m and 1334 m a.s.l. and the variance of the slopes is between 1 and 34°. Eighty-one of the plots are located in formally protected areas of different categories from water protection areas to strict reserves with different levels of restriction on active forest management.

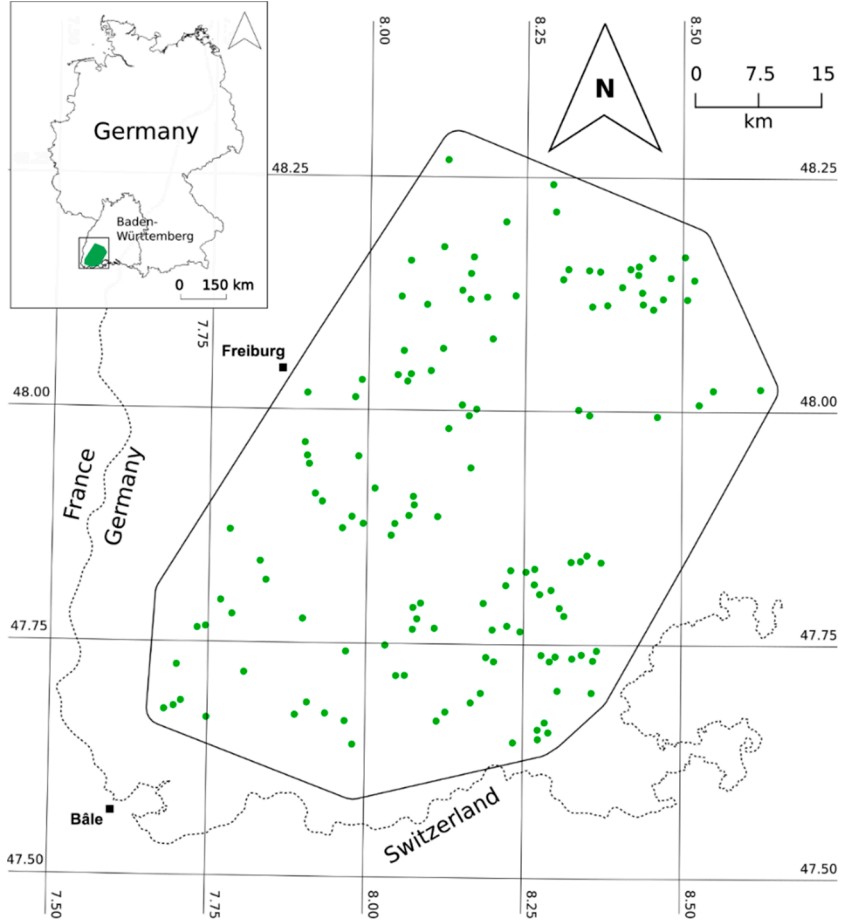

**Figure 1.** Map of the ConFoBi research area with research plots marked as green circles. The dotted line indicates the border of the state of Baden-Württemberg to France and Switzerland.

*2.2. Assessment of Tree-Related Microhabitats*

The field assessment of TreMs was carried out in leafless and snow-free periods between November 2016 and May 2017. The 15 trees with the largest crown area per plot were selected based on RS data prior to fieldwork. We focused on large trees, since literature has shown that larger trees bear significantly more TreMs compared to smaller ones [47,48] and we aim to select trees that provide a high quantity of these structures with the help of remote sensing. The selection of sample trees followed a stepwise approach. First, we automatically delineated individual tree crowns of all trees in all plots by applying the TreeVis software [49]. A digital surface model (DSM) was photogrammetrically generated from a combination of aerial images (40 cm ground sampling distance, (similar to [50]) and a digital terrain model (DTM) based on LiDAR flights. From the DSM, 15 large living trees per plot, based on the delineated crown areas, could be identified. The sample size of 15 trees is derived from the local retention forestry concept that is applied to all state forests in Baden-Württemberg [34]. In this concept, groups of 15 habitat trees are selected per three hectares as small retention sites. An assessment of all trees in the plots was not feasible for logistic reasons, but by selecting 15 large trees, we captured most

of the variation (80% based on a rarefaction analysis) of TreMs in the plots [44]. The tree species was not a selection criterion and it was not the goal to inventory the absolutely largest trees per plot, but to find a feasible solution for selecting large trees prior to the full inventory.

The TreM inventory was derived from Kraus et al. [51], which has been slightly adapted and is now commonly used throughout Europe [27]. The catalogue used for the inventory [51] included these TreMs (for detailed information as minimum sizes to be recorded see Table A1):

- Cavities: Woodpecker cavities, trunk and mould cavities, branch holes, dendrotelms as well as insect galleries and bore holes;
- Injuries and wounds: bark loss or exposed sapwood, exposed heartwood or trunk and crown breakage, cracks and scars
- Bark: space between bark and sapwood forming a shelter or pocket, coarse structure
- Deadwood: dead branches and limbs or crown deadwood
- Deformation and growth form: root buttress cavities, witches broom, cankers and burrs
- Epiphytes: fruiting bodies of fungi, myxomycetes, epiphytic crypto- and phanerogams
- Nests: nests of vertebrates and invertebrates
- Other: sap and resin run, micro soil

A simple handheld global navigation satellite system (GNSS) was used to locate the pre-selected sample trees in the field. For each sample tree, DBH, species and an inventory of TreMs were recorded. All TreMs on a tree were recorded with type and count. TreMs in the upper parts of trees, including canopy branches, were identified with binoculars. To prevent an observer effect, all inventories were carried out by the same team of two observers [45]. See supporting material (Table A1) for a full list of included TreMs. For the statistical analyses, TreM abundance is defined as the sum of all recorded TreMs of 15 large trees per plot. The richness of TreMs was calculated as the sum of all different TreMs of the inventoried 15 trees per plot.

*2.3. Acquisition of Data with the Unmanned-Aerial Vehicle (UAV)*

All research plots were inventoried with a multirotor UAV (OktoXL 6S12, Mikrokopter GmbH, Moormerland, Germany; Figure 2) carrying a consumer-grade full frame RGB camera (Alpha 7R, Sony Europe Limited, Weybridge, Surrey, UK) with global shutter a 35 mm prime lens. The flights were carried out in snowless conditions between March 2017 and April 2018. In order to minimize the data collection timeframe but still include all plots, light and weather conditions varied per flight. For each flight, the aircraft was set to "automatic mode", flying over the plots at 80 m above ground in a crisscross pattern using the onboard GNSS and compass for navigation following a preflight programmed flight plan (see [52] for details). The camera was aligned nadir and perpendicular to the flight direction and triggered automatically every 3–4 m by the drone, resulting in forward overlaps >95% and ground sampling distances (GSD) of about 1.1 cm. The camera was set to an exposure time of 1/2000 s, aperture F/2.8, the ISO-value was set on site based on the light conditions. Given that the aircraft only maneuvers according to its relative height to the starting point, we planned flights to begin at the lowest point of the plot, thus avoiding collisions with trees in steep terrain. Consequently, on-site assessments of feasible launch locations verified or altered the starting point defined in the preflight plan. As a result, flight heights were roughly stable within plots but occasionally varied between plots. The mean realized flight height was 96 m (SD: 19 m), which generated a sideward overlap between 83% and 91% and ground sampling distances that varied between one and two cm [52].

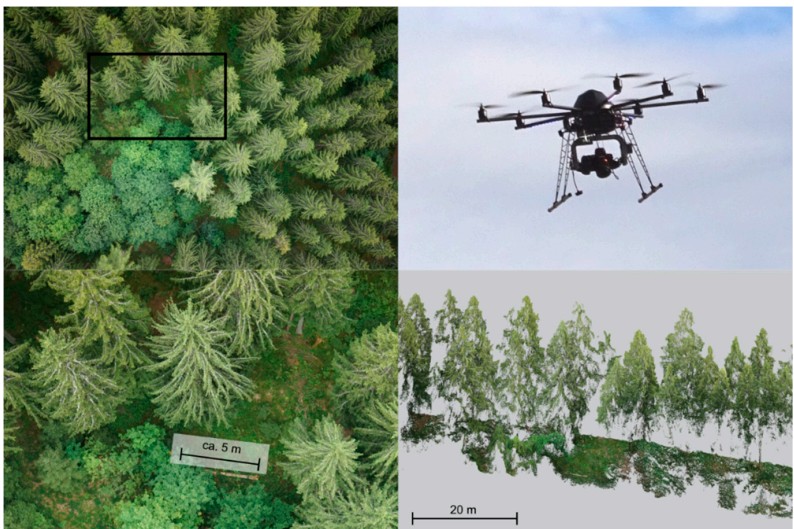

**Figure 2.** UAV image acquisition; the top-left panel shows a raw image of the UAV (**top right**). The **lower-left** panel shows a close-up of the same image (frame in **top left** image) with a log of ca. 5 m length. The **lower-right** panel illustrates a transect of the resulting point cloud.

## 2.4. Data Acquisition with a Terrestrial Laser Scanner (TLS)

Single scans were conducted between September 2017 and May 2018 at the center of every plot, which was marked with a metal pin using a real-time kinematic (RTK) GNSS. Each scan was carried out with a Faro Focus 3D 120 (Faro Technologies Inc., Lake Mary, FL, USA; Figure 3) terrestrial laser scanner set to 0.044° resolution (7.76 mm point distance at 10 m distance to scanner). A full 360° horizontal and 150° vertical angular range was covered, resulting in a maximum of 29 million points per scan. The scanner was placed on a tripod at 1.3 m above ground. Instrument heights, date and time, GPS-location and qualitative weather information were recorded as metadata for every scan using a field tablet. The scanner automatically corrected its tilt and rotation using internal sensors. See Figure 3 for an impression of the dataset quality.

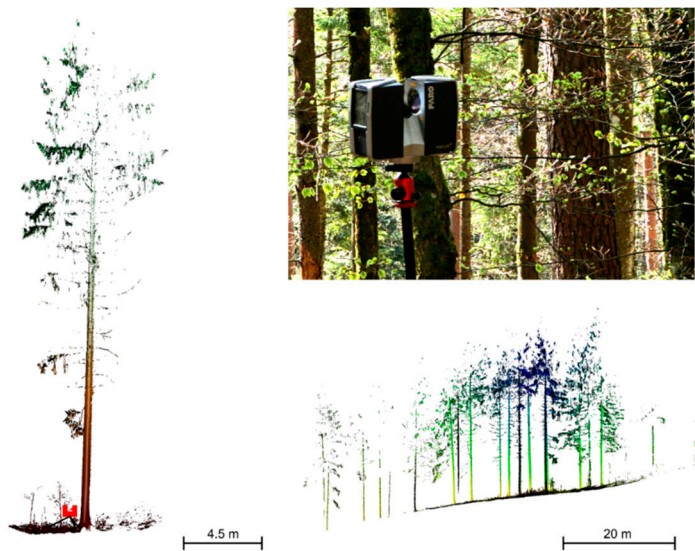

**Figure 3.** The left panel shows an example of dataset details for a single tree from TLS scanner (**top-right** panel), the scanner position is illustrated by the red icon. In a 10 m long transect through the scan (**lower right** panel), trunks are represented in high details whereas the depiction of crowns suffers from occlusion effects.

### 2.5. Processing of Data Obtained by the UAV

The RGB images of each plot (see Figure 2 for an example) were georeferenced by matching the timestamps from the camera and the onboard GNSS on the copter. Afterward a full structure from motion (SfM) workflow was performed using the commercial Agisoft Photoscan software (v. 1.3.4, 2017), including image matching, block adjustment, dense point cloud generation, digital surface model (DSM) and orthomosaic generation. The resolution of the raw images was lowered by a magnitude of four (1/16th of the original pixel count) before dense point cloud generation to save computing time (see Frey et al., 2018 for a detailed description of the processing).

The LidR package [53] in R [54] was used for further evaluation of point clouds. We used a DTM from previous state-wide LiDAR flights (LGL, 2000–2005) with 1 m resolution to normalize the terrain heights (lasnormalize-function) and clip (lasclip-function) the point cloud to the plot borders. Next, the lasmetrics-function was used to compute various summary statistics (Table 1).

The DSM of each plot was imported to a PostgreSQL database (v. 9.6; Group and others, 2011), clipped with the plot borders and normalized with the previously mentioned DTM. Summary statistics were computed using the PostGIS (v 2.3.3; [55]) functions st_summarystats, st_tri and st_quantile (Table 1).

### 2.6. Processing of Data Obtained by the TLS

The raw data from the scanner (Figure 3) was transferred to Faro Scene (v 6.2.4.30) and noise was removed by applying the standard outlier removal with default parameters. All further processing was done in R (version 3.5.0, [54]). After normalizing the point clouds using the previously mentioned DTM, we applied various summary statistics (Table 1) from the literature. These included indices based on 10 $cm^3$ voxels such as the Effective Number of Layers (*ENL*; [56]), and basic statistical measures to describe the point heights distribution derived from the DTM normalized point cloud.

An additional category of indices was used to quantify the complexity of distribution of points in 3D space like the Stand Structural Complexity Index (*SSCI*; [56]). This index is the ratio between the perimeter and area of a polygon constructed from a single cross section from the scanner as a measure of spatial complexity [4]. It averages over all cross sections (scan stripes) that the scanner measures during a full 360° scan. This dimensionless measure of complexity (MFRAC) is afterwards scaled using the natural logarithm of the ENL (Equation (1); [4]):

$$SSCI = MFRAC^{ln(ENL)} \qquad (1)$$

### 2.7. Terrain Information

A LiDAR-based DTM with 1 m resolution was available based on data from the responsible federal authorities [57]. From this DTM the average altitude of each plot was extracted. For every cell the slope was calculated and averaged over the plot. The aspect was calculated based on the four corners of the plot. All calculations were accomplished using PostGIS functions [55].

### 2.8. Statistical Analyses

Many common forest variables can be extracted from TLS-data [58,59], yet certain methods are only applicable to single-layer stands with relatively uniform tree distributions. The influence of common forest attributes on TreMs has been researched and the prediction based on this information is well established [44,47]. We build models including the derived RS parameters as predictors for TreM abundance and richness.



**Table 1.** Summary of the included remote sensing variables from UAV and TLS measurements included in the statistical analyses.

| Predictor | Description | Formula | Ref. |
|---|---|---|---|
| UAV-NDSM NDSM_Mean | Mean vegetation height | $NDSM_{mean} = mean(DSM - DTM)$ | |
| NDSM_SD | Standard deviance of the NDSM | | |
| NDSM_TRI | Mean NDSM terrain ruggedness index (mean difference of a central pixel to its surrounding 8 pixels) | | [60] |
| Gap_Share | Proportion of the area with vegetation lower than 2 m. | | [50] |
| UAV Pointcloud PD | Point-density measured in pt/m$^2$ | | |
| PR | Penetration-rate—share of points in the below 1 m strata. | | |
| UAV and TLS point cloud VCI | Vertical complexity index. Normalized Shannon index on points in 1 m height bins with a maximum height of 40 m | $VCI = (-\sum_{i=1}^{40} ([p_i * ln(p_i)]))/ln(40)$ | [61] |
| Z_Kurt | kurtosis of height distribution | | [53] |
| Z_Mean | mean height | | [53] |
| Z_Max | maximum height | | [53] |
| Z_SD | standard deviation of height distribution | | [53] |
| Z_Skew | skewness of height distribution | | [53] |
| zQ10 … zQ90 | Height quantiles of normalized point clouds in 40% steps | | |
| TLS point cloud MFRAC | Mean fractal dimension index from all cross sections (vertical scanning columns) of the TLS scan. Index includes perimeter (P) and area (A) of the cross sections. | $MFRAC = mean\left(\frac{2*ln(0.25*P)}{ln(A)}\right)$ | [4] |
| ENL | Effective number of layers describes the diversity between the forest strata using an inverse Simpson index and a voxel approach | $ENL = 1/\sum_{i=1}^{Ntop} p_i^2$ | [56] |
| SSCI | Stand structural complexity index combines MFRAC with ENL as scaling factor | $SSCI = MFRAC^{ln(ENL)}$ | [4] |
| Mean_Dist | Mean measurement distance of the scanner. | | |
| DTM Altitude | Mean, max, min plot altitude | | |
| Slope | Mean, max, min plot slope | | |
| Aspect | Plot orientation in divergence from north | | |

As a first step, we tested all predictors for collinearity using a correlation plot. Since several of the predictor variables are strongly collinear related we used a principal component analyses (PCA) to combine the predictors into a smaller set of independent components. We selected components to cover 90% of the variance of the original predictors. In the case this resulted in eleven principal components (PC), which were used as new predictor variables in the final statistical analyzes. With this setup, we followed the procedure described in Ciuti et al. [21]. To account for possible nonlinear relationships between the predictors and the response we used generalized additive models (GAMs). GAMs combine General Linear Models with smoothing splines, thereby allowing to fit the response curves as closely as possible to the data, within a permitted level of smoothing. The smoothing function avoids that the flexible model structure over-fits the data. GAMs with cubic splines and shrinkage (method = 'REML'), conducted in the R MGCV package were used to fit one model for TreM-richness with poisson error distribution and one for TreM-abundance with negative binomial error distribution (R version 3.5.0, [54]). To gain a better understanding of the factors that drive the abundance and richness of TreMs we correlated the original predictors with the PCs to see which had the most significant influence using the dimdesc function in the FactorMineR package in R [62]. We repeated the modelling steps for single sensor datasets (UAV or TLS) to verify that the models improve from the combinations of sensors. The predictive performance of the models for abundance and richness including both sensors was additionally checked using a 1000 fold cross validation, while leaving $\frac{1}{4}$ of the data as test dataset out each time. Models were fitted excluding the test data (training dataset) and we predicted the response values for training and test dataset separately and compared the root mean squared errors (RMSE) over the 1000 repetitions using a *t*-test.

## 3. Results

We were able to efficiently reduce the number of predictor variables using the PCA from 28 to eleven predictors while still covering more than 90% of the explained variance of the original predictors according to the PCA. The resulting models described 44.3% of the deviance of TreM abundance ($R^2$: 0.31) and 37.8% of the richness ($R^2$: 0.31). Five of the PCs had a significant influence on the TreM abundance and four on the richness. These are described in Table 2 in more detail.

The significant PCs cover a wide range of variables including those describing the height variations of the point cloud and therefore the horizontal layer complexity (PC2), the canopy complexity and height (PC3), the terrain slope (PC4), the shape complexity (PC6) as well as the terrain altitude (PC10), according to the correlations between the variables and the PCs (Table 2).

Results of the prediction compared with the observed TreMs (Figure 4) show that the prediction underestimates the abundances and richness of TreMs in plots which provide greater numbers of TreMs. In contrast, in plots with few TreMs the model overestimates slightly. There are less observations of plots with a high and a low abundance and richness of TreMs compared to the medium level of TreM provisioning.

**Table 2.** Statistical significant principal components (*p*-values: * < 0.05, ** < 0.01, *** < 0.001) according to the GAMs and their most important influencing variables based on the correlation between the PC and the predictor calculated by the dimdesc function from the factominer package [62].

| Principal Component | Significant For | Most Influencing Predictor Variables | (Correlation) | Description |
|---|---|---|---|---|
| **PC2** | Abundance *** | UAV Z SD | (0.87) | Horizontal layer complexity |
| | | UAV NDSM SD | (0.77) | |
| | | TLS mean dist. | (−0.28) | |
| | Richness *** | UAV zQ10 | (−0.61) | |
| **PC3** | Abundance * | UAV Gap share | (0.56) | Canopy complexity & canopy height |
| | | UAV PR | (0.43) | |
| | | UAV z Mean | (−0.51) | |
| | | NDSM Mean | (−0.60) | |
| **PC4** | Abundance * | Max Slope | (0.68) | Slope & point cloud density |
| | | Min Slope | (0.63) | |
| | Richness *** | TLS zQ10 | (−0.33) | |
| | | AV PD | (−0.72) | |
| PC6 | Abundance ** | UAV PR | (0.44) | Shape complexity |
| | | TLS SSCI | (0.39) | |
| | Richness ** | Max Slope | (−0.40) | |
| | | Min. Slope | (−0.42) | |
| **PC10** | Abundance *** | Avg Altitude | (0.51) | Terrain altitude & slope |
| | | UAV VCI | (0.35) | |
| | Richness * | UAV TRI | (−0.23) | |
| | | Min Slope | (−0.33) | |

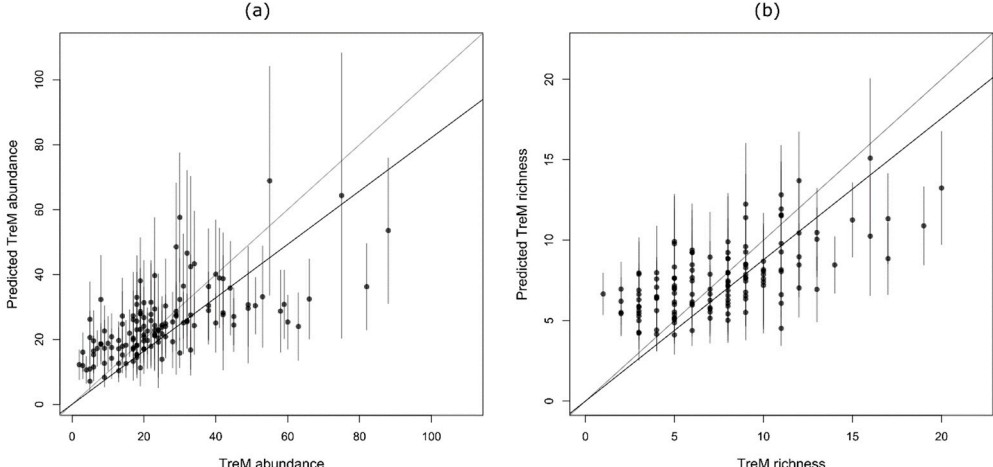

**Figure 4.** Observed vs. predicted abundance (**a**) and richness (**b**) of tree-related microhabitats. The predicted TreM abundance and richness (y-axes) is compared with the observed one (x-axes). The 1:1 lines are displayed in light grey, while the trend-lines are black and have a fixed intercept of 0. The whiskers show the 95% confidence interval of the prediction.

The cross validation showed no significant differences between the RMSEs of the training and test datasets neither for abundance (*p*: 0.23) nor for richness (*p*: 0.51). Figure 5 shows the mean RMSE over the number of cross validations.

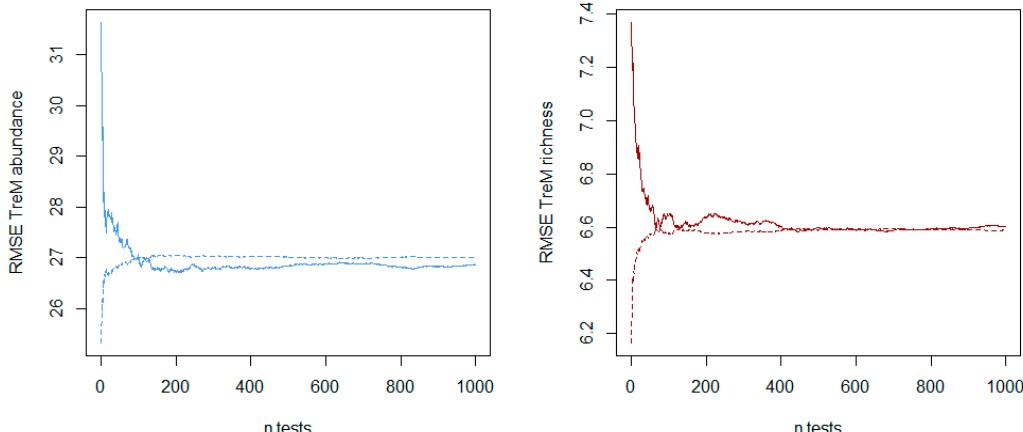

**Figure 5.** Results of the cross validation. Straight lines indicate the mean RMSE of the training data and dotted lines of the test data as cumulative mean up to the specific test (all RMSEs averaged until *n*-test). The left panel shows the results for TreM abundance, the right panel for TreM richness.

The model that contained only data collected by UAV performed a little better for the TreM abundance prediction (+6% deviance explained), while the model that contained data collected by TLS predicted the TreM richness slightly better (+4% deviance explained). The models containing both sensors explained on average 4% more deviance than the single sensor datasets.

## 4. Discussion

The models were able to explain between 44% and 38% of the deviance in abundance and richness of tree-related microhabitats respectively based on close-range remote sensing parameters. The selected remote sensing data, therefore, allows a moderate prediction of the occurrence of TreMs in forests. This analysis helps to clarify which forest structural variables derived from remote sensing are related to TreMs. Through our results guidance for the selection of areas with high quality habitat trees based on measures of forest structure derived from close-range remote sensing can be established.

Forest structure could be described by two dimensions in our models taken from the PCA that significantly predicted the abundance and richness of TreMs. One of the dimensions could be summarized as indication of canopy complexity and height expressed by PC3 which relate to gaps in the canopy. It is known that gaps are related to forest biodiversity [63] and the size influences forest dwelling species, for instance ants [64] which are in turn related to TreMs. In addition remote sensing techniques have been used to detect habitat thresholds for indicator species as the three-toed woodpecker (*Picoides tridactylus*) based on canopy properties such as the amount of deadwood crown size [19], which will eventually result in canopy gaps and points in a similar direction as our results. Therefore, we could suggest to focus selection activities of habitat trees to forest areas with a certain degree of gaps in the canopy. Similarly, a second dimension of the PCA (PC2) could be summarized as a description of the horizontal layer complexity. One group of TreMs that is relatively great in abundance in our study area and particularly influenced by the canopy structure are epiphytes [65,66]. This might to some extent reflect why the canopy structure influences both the abundance and richness of TreMs.

From a more technical point of view, the penetration rate and shape complexity expressed by the SSCI are as well significant in the prediction of the abundance and richness of TreMs (PC6). Here, we might refer to the importance of large trees per se that shape the structure of the forest [67,68] and especially relate to the extent of buttressing as well as the occurrence of cavities. Another reason why shape complexity recorded by remote sensing and expressed as the SSCI index influences the abundance and richness of TreMs might refer to tree species richness [56]. Forests with a more diverse tree species composition commonly show a more complex structure, especially if the applied

silvicultural system is close-to-nature forest management [46]. Thus, the factor explaining the relation of shape complexity and TreMs might be a well-known complementary effect of tree species mixtures related to forest type, an effect especially meaningful for TreM richness [44,69]. This includes the fact that forest types including broadleaved trees provide a higher abundance and richness of TreMs compared to less complex coniferous forest types [44,69]. The influence of mixtures including especially broadleaf tree species, and in our case larger shares of beech, increase the abundance and richness of TreMs as well as the shape complexity of the inventoried forests [44,47].

Other factors that were included in the axes of the PCA as significant predictors refer to geographic particularities of the study area in the Black Forest. It has been shown that altitude measured in the field increases the number of TreMs per plot [44], this similarly and not surprisingly holds true for the altitude derived from RS (PC10). We assume that this increase of TreMs in higher elevations is related to disturbances that differ from lower elevations such as longer periods of snow cover, with snow and substrate movements creating injuries at lower stem sections as well as differences in substrate e.g., less humus that allows the formation of a greater number of buttress cavities [44]. In addition to altitude, the slope of the plots was included as a principal component that significantly predicts the abundance and richness of TreMs (PC4). On steeper slopes, less intense management took place due to harvesting difficulties [70], hence more broadleaved trees remained compared to less steep and intensively managed areas dominated by the main commercial coniferous species Norway spruce and silver fir. Therefore, more TreMs can be found on steeper slopes as trees with TreMs that could be considered as "defects" for timber production are not removed as rigorous as in less steep terrain.

The relatively low importance of stand height to TreM abundance was surprising, since the occurrence of TreMs has been shown to correlate with forest age and DBH, which is correlated with the stand height [47]. Two factors might address this result, both related to the study design. First, the TreM sampling approach was based on the 15 trees with the largest crown radius, instead of a full census of all trees in the plots. In the majority of instances, these large living trees were in the top canopy layer and thus had a similar height distribution per plot. Therefore, the overall vegetation height in our plots was relatively uniform and thus not a strong descriptor of TreM abundance and richness. Secondly, the plots in our data set are relatively homogenous: the project design selected plots with tree ages above 60 years, and many of the plots were previously managed. We expect vegetation height being a better descriptor for TreMs in more variable forest structures since there is a positive relationship between DBH and TreM abundance and richness [47,71,72].

A further limitation of the study design was that the TLS sampling used only one scan at the center of the research plot and did not cover the full area of the plot, due to heavy occlusion effects [73]. A design with multiple scan locations might have covered the plot better, however Ehbrecht et al. [56] have shown for the ENL index as structural descriptor, that single scans are representative for a stand, which should be valid for other stand characteristics as well. Single scans require far less effort than multiple scans. This holds not just true during the scanning phase, but as well in the following processing steps, as the labor-intensive target placement and matching is not required. It might still be very helpful in the future and a further step towards a more efficient and objective selection of habitat trees, to scan individual trees on a 360° angle. This might as well allow a full detection of TreMs on each individual tree, which was not the research aim of this particular study. In situations with very dense undergrowth matching might not even been feasible. These points apply as well to the TreM inventory, since sampling all trees would be extremely time consuming. Earlier studies have shown that the 15 largest trees cover most of the variance of TreM richness and abundance (80%) in the plot [44]. The UAV-SfM dataset is easily recordable for the whole plot, but the positioning was suboptimal, which makes all the datasets only comparable to a limited extend. Nevertheless, all applied techniques sample the stand in a specific manner for a certain goal and create a limited, but representative model of the stand with an acceptable effort. The great number of plots and their distribution in the landscape required a very efficient sampling design, which is time-effective as well. Further advances in sensor technologies with very dense aerial or UAV LiDAR might overcome these shortcomings of incomplete

representation of the geometry of the stand and make new structural indices or the full detection of TreMs possible [38,74]. The sampling effort of the different methods (manual inventories, UAV, mobile or terrestrial RS) differs strongly. While currently terrestrial and manual inventories might take multiple hours per ha, UAV systems can cover a similar area within 10–20 min. Mobile scanning systems and advances in UAV technology might lower the sampling effort even further [74].

Another limitation of our study, which may explain the weak link between some measurements of forest complexity or structure, was the absence of old-growth forests in our research site. Despite the inclusion of some strict reserves and other protected areas in the study, most plots are located in forests that are managed or where the structure is still strongly influenced by previous management. It is possible that old-growth forests will show a stronger link between the present RS indices and TreM assessments for structural elements affecting forest biodiversity [39,43].

While different RS- and TreM-based studies have shown promising results for the quantification of diversity of different taxa [1,3,13,38], their potential as combined descriptors of biodiversity had not yet been researched. As technical progress advances, new options for the detection of particular TreMs at very fine scales will become available [38], and information on the most applicable set of predictors can help to identify the most objective, cost and time efficient inventory methods for the selection of key retention elements as habitat trees for forest biodiversity.

## 5. Conclusions

This study detected several relationships between measures of structural complexity and the abundance and richness of TreMs in the southern Black Forest. Most notably, structural indices that can be derived from data collected by combination of UAV and TLS, are related to the abundance and richness of TreMs. This supports our ecological understanding of structural complexity as significant driver of the provisioning of TreMs at the plot-level. The RS techniques offer a complementary approach for identifying relevant predictors of forest structures that provide a high abundance and richness of TreMs and thus facilitate the selection of retention elements such as habitat trees beyond the level of single-species information. None of the proposed models alone might be able to predict TreMs sufficiently for a habitat tree selection based on RS only, but our results offer new evidence for forest biodiversity conservation. This might for instance apply to a pre-selection of areas of retention interest, where habitat trees can be found in greater numbers and quality based on the prediction of a higher abundance and richness of TreMs. In these patches that offer a more complex fine-scale forest structure described by RS, more individuals suitable as habitat trees might be found.

**Author Contributions:** The authors T.A. and J.F. contributed evenly to the data acquisition, formal analyses and writing and revising of the draft. Conceptualization, J.F., T.A. and J.B.; methodology, J.F. and T.A.; software, J.F.; validation, T.A., J.F. and J.B.; formal analysis, J.F.; investigation, J.F. and T.A.; resources, J.B., J.F. and T.A.; data curation, J.F. and T.A.; writing—original draft preparation, J.F. and T.A.; writing—review and editing, J.B.; visualization, J.F.; supervision, J.B.; project administration, J.B.; funding acquisition, J.B. All authors have read and agreed to the published version of the manuscript.

**Funding:** This study was funded by the German Research Foundation (DFG), ConFoBi project number GRK 2123.

**Acknowledgments:** The authors would especially like to thank Taylor Shaw for the intensive proofreading as a native speaker and the numerous text improvements. We also thank the whole ConFoBi team especially Barbara Koch, who helped with her constructive comments to improve this manuscript. We thank the various public forest authorities involved for facilitating this research and the state agency of spatial information and rural development of Baden-Württemberg (LGL) for the provisioning of data.

**Conflicts of Interest:** The authors declare no conflict of interest.

## Appendix A

**Table A1.** Microhabitat type, detailed description and number of records from microhabitat inventory.

| Microhabitat Type | Code | Description | N |
|---|---|---|---|
| Bark | BA11 | Bark shelter, open bottom | 26 |
| Bark | BA12 | Bark pocket, open top | 10 |
| Woodpecker cavity | CV11 | Cavity entrance about ø = 4 cm | 2 |
| Woodpecker cavity | CV12 | Cavity entrance about ø = 5–6 cm w | 9 |
| Woodpecker cavity | CV13 | ø > 10 cm Woodpecker hole in the trunk | 6 |
| Woodpecker cavity | CV14 | ø ≥ 10 cm feeding hole | 13 |
| Woodpecker cavity | CV15 | Woodpecker "flute"/cavity string | 6 |
| Trunk/ mould cavity | CV21 | ø ≥ 10 cm (ground contact) | 14 |
| Trunk/mould cavity | CV22 | ø ≥ 30 cm (ground contact) | 13 |
| Trunk/mould cavity | CV23 | ø ≥ 10 cm (no ground contact) | 21 |
| Trunk/mould cavity | CV24 | ø ≥ 30 cm (no ground contact) | 11 |
| Trunk/mould cavity | CV25 | ø ≥ 30 cm/semi-open | 4 |
| Trunk/mould cavity | CV26 | ø ≥ 30 cm/open top | 0 |
| Branch hole | CV32 | ø ≥ 10 cm holes from breakage | 39 |
| Branch hole | CV33 | Hollow branch, ø ≥ 10 cm | 133 |
| Dendrotelm | CV42 | ø ≥ 15 cm/trunk base | 11 |
| Dendrotelm | CV44 | ø ≥ 15 cm/crown | 6 |
| Insect gallery/bore holes | CV51 | Gallery with single small bore holes | 4 |
| Insect gallery/bore holes | CV52 | Large bore hole | 1 |
| Dead branch | DE11 | ø 10–20 cm, ≥ 50 cm, sun exposed | 125 |
| Dead branch | DE12 | ø > 20 cm, ≥ 50 cm, sun exposed | 7 |
| Dead branch | DE13 | ø 10–20 cm, ≥ 50 cm, not sun exposed | 231 |
| Dead branch | DE14 | ø > 20 cm, ≥ 50 cm, not sun exposed | 21 |
| Dead branch | DE15 | Dead top ø ≥ 10 cm | 12 |
| Fungi fruiting body | EP11 | Annual polypores, ø > 5 cm | 2 |
| Fungi fruiting body | EP12 | Perennial polypores, ø > 10 cm | 4 |
| Fungi fruiting body | EP13 | Pulpy agaric, ø > 5 cm | 4 |
| Fungi fruiting body | EP14 | Large ascomycetes, ø > 5 cm | 0 |
| Myxomycetes | EP21 | Myxomycetes, ø > 5 cm | 1 |
| Epiphyte | EP31 | Epiphytic bryophytes, >25% trunk | 311 |
| Epiphyte | EP32 | Epiphytic foliose/lichens, >25% trunk | 387 |
| Epiphyte | EP33 | Lianas, coverage >25%, | 16 |
| Epiphyte | EP34 | Epiphytic ferns, >5 fronds | 5 |
| Epiphyte | EP35 | Mistletoe in tree crown | 275 |
| Root buttress cavity | GR12 | ø ≥ 10 cm, natural cavity | 956 |
| Root buttress cavity | GR13 | Trunk cleavage, length ≥ 30 cm | 11 |
| Witches broom | GR21 | Witches broom, ø > 50 cm | 72 |
| Witches broom | GR22 | Water sprout, dense epicormics | 6 |
| Canker or burr | GR31 | Cancerous growth, ø > 20 cm | 14 |
| Canker and burr | GR32 | Decayed canker, ø > 20 cm | 25 |
| Bark loss | IN 11 | Bark loss 25–600 cm$^2$, decay stage < 3 | 255 |
| Bark loss | IN12 | Bark loss > 600 cm$^2$, decay stage < 3 | 63 |
| Bark loss | IN13 | Bark loss 25–600 cm$^2$, decay stage = 3 | 24 |
| Bark loss | IN14 | Bark loss > 600 cm$^2$, decay stage = 3 | 23 |
| Exposed heartwood | IN21 | Broken trunk, ø ≥ 20 cm at broken end | 5 |
| Exposed heartwood | IN22 | Broken tree crown/fork | 11 |
| Exposed heartwood | IN23 | Broken limb, ø ≥ 20 cm at broken end | 19 |
| Exposed heartwood | IN24 | Splintered stem, ø ≥ 20 cm | 0 |
| Crack or scar | IN31 | Length ≥ 30 cm | 15 |
| Crack or scar | IN32 | Length ≥ 100 cm | 13 |
| Crack or scar | IN33 | Lightning scar | 2 |
| Crack or scar | IN34 | Fire scar, ≥600 cm$^2$ | 0 |
| Nest | NE11 | Large vertebrate nest, ø > 80 cm | 2 |
| Nest | NE12 | Small vertebrate nest, ø > 10 cm | 40 |
| Nest | NE21 | Invertebrate nests in trunk | 0 |
| Sap and resin run | OT11 | Sap flow, >50 cm, fresh, deciduous | 0 |
| Sap and resin run | OT12 | Resin flow/pockets, >50 cm, coniferous | 542 |
| Micro soil | OT21 | Crown micro soil | 9 |
| Micro soil | OT22 | Bark micro soil | 11 |

**Table A2.** Overview of the plots inventoried including DBH, microhabitat abundance and richness, altitude, management type as well as forest type.

| Plot | Mean DBH (cm) (SD) | Microhabitat Abundance | Microhabitat Richness | Altitude (m) | Management | Forest Type |
|------|--------------------|------------------------|-----------------------|--------------|------------|-------------|
| 1 | 66.1 (19.7) | 99 | 21 | 1247 | Strict-protection | Coniferous-broadleaved |
| 2 | 53.1 (20.8) | 49 | 16 | 873 | Uneven-aged | Mixed-coniferous |
| 3 | 66.7 (17.2) | 73 | 19 | 1226 | Strict-protection | Coniferous-broadleaved |
| 5 | 69.5 (21.3) | 44 | 14 | 806 | Even-aged | Coniferous-broadleaved |
| 7 | 43.5 (13.2) | 47 | 14 | 1334 | Strict-protection | Coniferous-broadleaved |
| 8 | 42.6 (10.1) | 35 | 7 | 1295 | Mixed Management | Coniferous-broadleaved |
| 9 | 55.4 (12.9) | 34 | 10 | 716 | Even-aged | Coniferous-broadleaved |
| 10 | 69.9 (16.4) | 45 | 11 | 713 | Strict-protection | Coniferous-broadleaved |
| 11 | 50.8 (9.4) | 25 | 8 | 904 | Mixed Management | Mixed-coniferous |
| 14 | 57.8 (13.6) | 21 | 10 | 512 | Even-aged | Coniferous-broadleaved |
| 15 | 70.6 (11.5) | 59 | 13 | 1069 | Even-aged | Coniferous-broadleaved |
| 16 | 82.2 (23.2) | 141 | 23 | 947 | Even-aged | Coniferous-broadleaved |
| 17 | 61.4 (9.1) | 71 | 9 | 1069 | Even-aged | Pure-coniferous |
| 18 | 69.5 (14.3) | 72 | 6 | 947 | Even-aged | Mixed-coniferous |
| 19 | 57.2 (11.4) | 72 | 16 | 1014 | Even-aged | Coniferous-broadleaved |
| 20 | 59.6 (9.9) | 52 | 8 | 992 | Even-aged | Mixed-coniferous |
| 21 | 52.2 (10.8) | 53 | 11 | 1088 | Even-aged | Coniferous-broadleaved |
| 22 | 48.7 (10.6) | 17 | 6 | 715 | Even-aged | Coniferous-broadleaved |
| 28 | 70.0 (11.6) | 53 | 16 | 1026 | Even-aged | Coniferous-broadleaved |
| 30 | 58.6 (9.4) | 11 | 4 | 510 | Even-aged | Coniferous-broadleaved |
| 31 | 43.6 (7.2) | 21 | 10 | 541 | Even-aged | Coniferous-broadleaved |
| 33 | 53.5 (13.7) | 39 | 9 | 985 | Even-aged | Mixed-coniferous |
| 34 | 43.3 (7.2) | 32 | 7 | 928 | Even-aged | Pure-coniferous |
| 35 | 54.4 (5.5) | 66 | 9 | 533 | Even-aged | Coniferous-broadleaved |
| 36 | 44.9 (7.3) | 34 | 6 | 1050 | Even-aged | Pure-coniferous |
| 37 | 58.1 (8.2) | 81 | 13 | 1056 | Even-aged | Coniferous-broadleaved |
| 38 | 49.2 (14.2) | 30 | 9 | 904 | Even-aged | Mixed-coniferous |
| 39 | 66.3 (15.6) | 76 | 13 | 649 | Even-aged | Coniferous-broadleaved |
| 44 | 61.3 (7.8) | 35 | 10 | 835 | Mixed Management | Coniferous-broadleaved |
| 45 | 54.0 (10.8) | 34 | 8 | 587 | Even-aged | Coniferous-broadleaved |
| 47 | 53.9 (19.3) | 73 | 15 | 744 | Even-aged | Coniferous-broadleaved |
| 48 | 54.7 (17.3) | 52 | 13 | 704 | Even-aged | Coniferous-broadleaved |
| 50 | 77.9 (18.5) | 86 | 13 | 775 | Uneven-aged | Mixed-coniferous |
| 53 | 64.2 (12.9) | 36 | 6 | 950 | Uneven-aged | Mixed-coniferous |
| 54 | 44.3 (21.1) | 16 | 11 | 734 | Even-aged | Coniferous-broadleaved |
| 55 | 58.4 (11.7) | 32 | 10 | 767 | Mixed Management | Coniferous-broadleaved |
| 56 | 53.1 (7.5) | 28 | 11 | 443 | Mixed Management | Coniferous-broadleaved |
| 57 | 58.7 (9.0) | 41 | 10 | 640 | Even-aged | Coniferous-broadleaved |

**Table A2.** *Cont.*

| Plot | Mean DBH (cm) (SD) | Microhabitat Abundance | Microhabitat Richness | Altitude (m) | Management | Forest Type |
|------|--------------------|------------------------|-----------------------|--------------|------------|-------------|
| 58 | 40.4 (20.9) | 20 | 11 | 694 | Mixed Management | Coniferous-broadleaved |
| 59 | 36.9 (10.0) | 16 | 6 | 634 | Even-aged | Mixed-coniferous |
| 60 | 56.9 (21.9) | 29 | 13 | 613 | Even-aged | Mixed-coniferous |
| 61 | 50.6 (8.0) | 24 | 6 | 515 | Even-aged | Coniferous-broadleaved |
| 63 | 56.7 (14.5) | 37 | 9 | 566 | Even-aged | Coniferous-broadleaved |
| 64 | 48.8 (14.7) | 49 | 15 | 717 | Even-aged | Coniferous-broadleaved |
| 65 | 40.4 (16.8) | 12 | 6 | 684 | Even-aged | Mixed-coniferous |
| 67 | 49.4 (11.1) | 18 | 7 | 740 | Even-aged | Mixed-coniferous |
| 68 | 40.4 (18.3) | 5 | 3 | 792 | Even-aged | Coniferous-broadleaved |
| 69 | 64.4 (15.9) | 43 | 10 | 794 | Even-aged | Mixed-coniferous |
| 71 | 45.7 (6.4) | 18 | 5 | 678 | Even-aged | Mixed-coniferous |
| 72 | 51.6 (15.0) | 25 | 4 | 713 | Even-aged | Mixed-coniferous |
| 73 | 62.5 (18.0) | 35 | 10 | 871 | Uneven-aged | Coniferous-broadleaved |
| 75 | 45.5 (11.6) | 36 | 13 | 885 | Even-aged | Coniferous-broadleaved |
| 76 | 53.3 (7.4) | 39 | 13 | 504 | Even-aged | Coniferous-broadleaved |
| 77 | 39.9 (6.1) | 17 | 5 | 778 | Even-aged | Mixed-coniferous |
| 78 | 61.7 (19.8) | 67 | 20 | 697 | Mixed Management | Coniferous-broadleaved |
| 79 | 63.2 (13.4) | 64 | 16 | 922 | Even-aged | Mixed-coniferous |
| 83 | 48.6 (5.4) | 61 | 10 | 971 | Even-aged | Coniferous-broadleaved |
| 84 | 71.3 (12.8) | 45 | 11 | 754 | Even-aged | Mixed-coniferous |
| 85 | 53.5 (13.8) | 28 | 12 | 769 | Even-aged | Mixed-coniferous |
| 86 | 45.2 (4.2) | 24 | 5 | 713 | Even-aged | Coniferous-broadleaved |
| 87 | 49.3 (10.9) | 76 | 6 | 1018 | Even-aged | Coniferous-broadleaved |
| 89 | 74.9 (10.4) | 28 | 9 | 701 | Even-aged | Mixed-coniferous |
| 91 | 64.9 (17.7) | 72 | 15 | 1082 | Even-aged | Coniferous-broadleaved |
| 93 | 64.8 (16.0) | 46 | 14 | 665 | Strict-protection | Coniferous-broadleaved |
| 94 | 47.8 (14.9) | 27 | 12 | 1000 | Even-aged | Coniferous-broadleaved |
| 96 | 45.3 (9.0) | 33 | 10 | 750 | Uneven-aged | Coniferous-broadleaved |
| 98 | 54.9 (8.2) | 60 | 9 | 1120 | Uneven-aged | Mixed-coniferous |
| 101 | 74.7 (13.6) | 40 | 10 | 986 | Uneven-aged | Mixed-coniferous |
| 102 | 52.3 (14.2) | 16 | 6 | 877 | Even-aged | Mixed-coniferous |
| 103 | 41.3 (9.9) | 17 | 6 | 841 | Even-aged | Coniferous-broadleaved |
| 104 | 49.1 (12.5) | 37 | 14 | 580 | Mixed Management | Coniferous-broadleaved |
| 105 | 55.5 (9.0) | 26 | 10 | 833 | Even-aged | Coniferous-broadleaved |
| 106 | 52.4 (16.8) | 27 | 10 | 774 | Even-aged | Coniferous-broadleaved |
| 107 | 53.0 (16.8) | 38 | 10 | 733 | Even-aged | Mixed-coniferous |
| 108 | 53.3 (10.2) | 25 | 8 | 1126 | Uneven-aged | Mixed-coniferous |

**Table A2.** *Cont.*

| Plot | Mean DBH (cm) (SD) | Microhabitat Abundance | Microhabitat Richness | Altitude (m) | Management | Forest Type |
|------|--------------------|------------------------|-----------------------|--------------|------------|-------------|
| 109 | 63.5 (7.3) | 47 | 12 | 888 | Uneven-aged | Coniferous-broadleaved |
| 110 | 43.3 (11.4) | 36 | 7 | 930 | Even-aged | Coniferous-broadleaved |
| 111 | 60.6 (15.2) | 58 | 12 | 682 | Even-aged | Coniferous-broadleaved |
| 113 | 41.9 (7.4) | 32 | 4 | 1160 | Even-aged | Coniferous-broadleaved |
| 114 | 76.6 (11.6) | 48 | 12 | 516 | Even-aged | Mixed-coniferous |
| 117 | 52.8 (14.3) | 23 | 13 | 857 | Even-aged | Mixed-coniferous |
| 118 | 76.8 (11.8) | 77 | 19 | 657 | Uneven-aged | Coniferous-broadleaved |
| 119 | 67.2 (17.6) | 48 | 16 | 887 | Uneven-aged | Coniferous-broadleaved |
| 121 | 56.2 (7.7) | 29 | 12 | 632 | Even-aged | Coniferous-broadleaved |
| 122 | 56.0 (14.7) | 30 | 13 | 527 | Even-aged | Coniferous-broadleaved |
| 123 | 53.4 (6.8) | 38 | 9 | 646 | Even-aged | Mixed-coniferous |
| 124 | 52.6 (12.8) | 35 | 10 | 929 | Mixed Management | Coniferous-broadleaved |
| 125 | 48.5 (13.2) | 31 | 12 | 533 | Even-aged | Coniferous-broadleaved |
| 127 | 59.8 (9.6) | 23 | 8 | 516 | Even-aged | Mixed-coniferous |
| 128 | 53.9 (12.1) | 63 | 18 | 982 | Even-aged | Coniferous-broadleaved |
| 129 | 69.1 (12.8) | 88 | 22 | 549 | Even-aged | Coniferous-broadleaved |
| 130 | 59.5 (11.2) | 60 | 17 | 978 | Even-aged | Coniferous-broadleaved |
| 131 | 59.7 (12.0) | 84 | 10 | 1033 | Even-aged | Pure-coniferous |
| 132 | 45.6 (5.7) | 13 | 6 | 862 | Mixed Management | Mixed-coniferous |
| 133 | 63.8 (10.0) | 80 | 23 | 743 | Mixed Management | Coniferous-broadleaved |
| 134 | 53.5 (12.2) | 18 | 8 | 898 | Even-aged | Mixed-coniferous |
| 135 | 44.2 (7.0) | 22 | 7 | 569 | Uneven-aged | Mixed-coniferous |
| 137 | 66.5 (11.0) | 25 | 7 | 815 | Uneven-aged | Mixed-coniferous |
| 138 | 55.8 (7.3) | 34 | 6 | 853 | Uneven-aged | Mixed-coniferous |
| 140 | 49.7 (14.4) | 8 | 5 | 744 | Even-aged | Mixed-coniferous |
| 148 | 54.9 (23.0) | 29 | 7 | 831 | Even-aged | Coniferous-broadleaved |
| 151 | 60.9 (10.2) | 20 | 5 | 851 | Even-aged | Mixed-coniferous |
| 156 | 54.8 (12.9) | 29 | 9 | 797 | Even-aged | Coniferous-broadleaved |
| 165 | 44.9 (12.1) | 28 | 8 | 924 | Even-aged | Coniferous-broadleaved |
| 167 | 42.2 (7.1) | 12 | 3 | 813 | Even-aged | Mixed-coniferous |
| 176 | 48.6 (7.7) | 27 | 7 | 749 | Even-aged | Mixed-coniferous |
| 177 | 47.4 (6.8) | 29 | 6 | 972 | Even-aged | Pure-coniferous |
| 178 | 49.9 (25.7) | 36 | 14 | 663 | Strict-protection | Coniferous-broadleaved |
| 179 | 50.3 (11.0) | 23 | 10 | 1003 | Even-aged | Mixed-coniferous |
| 181 | 58.9 (16.0) | 31 | 7 | 903 | Mixed Management | Coniferous-broadleaved |
| 186 | 38.0 (8.1) | 22 | 11 | 787 | Even-aged | Coniferous-broadleaved |

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
