# Peer review of "Predicting Tree-Related Microhabitats by Multisensor Close-Range Remote Sensing Structural Parameters for the Selection of Retention Elements"

_remotesensing, doi:10.3390/rs12050867_

Round 1

Reviewer 1 Report

Pls, find enclosed file!

Author Response

Dear reviewer,

Thank you very much for your positive and constructive feedback. Please find our replies to your comments in the attached document.

Best regards,

Julian Frey

Reviewer 2 Report

Hi,

The good part is that english is good. However, I have a few comments with regard to the experiments.

  1. The ConFoBi research plots are not evenly distributed. Consequently, one part of research area is better represented than the other . So the final conclusions can be only pertaining to those better represented regions and hence cannot be generalized. The experiment has to be altered to include these variations
  2. According to your abstract, the corr. coeff. between the RS predictors and TreMs is only 0.31, which is very low to be considered this approach a viable alternative to the existing approach. 
  3. Further, when proposing this approach, there is no comparison with the existing methods on their accuracy. 
  4. Table 2 gives the details of your Principal Components. 
    1. What are those R2 correspond to? 
    2. How are their related to the one in your abstract and conclusion?
    3. PCA transforms your features to Principal Components, which are some combination of your input features. While it may be possible for you to revert from PCA to original features, it is not possible to identify which feature are part of which components? How did you achieve it? 
  5. The surface is unlikely to be straight without variations. How would it affect your results? No understanding is provided on the effect of surface variations.
  6. Similarly, due to atmospheric conditions, the UAV is unlikely to be without any perturbations. This is expected to induce errors in observations. How are they overcome/corrected? Since, the sensors in the UAV will reorient it after a perturbation, the image acquired during a perturbation needs to be corrected as well.
  7. There are different species of trees, how their composition differ between the plots? 
  8. And the sample of 15 trees per plot belong to the same species or is there a distribution?

In simple terms, there are a lot of questions unanswered in this manuscript. Also, more experiments might be required. 

Hope this helps,

Author Response

Dear reviewer,

Thank you very much for your extensive and relevant feedback. Please find our replies to your comments in the attached document.

Best regards,

Julian Frey

Reviewer 3 Report

Dear editor and authors,

This manuscript presents a study on using parameters derived from optical imagery obtained with an UAV as well as terrestrial laser scanning to estimate forest habitat quality (based on TreMs, i.e. tree-related microhabitats). The manuscript is overall well-written, and pleasant to read. The topic is also worthy of investigation and could be of practical use for forest habitat monitoring. Prior to publication, however, I would suggest one methodological improvement or addition that would substantially increase insights into the actual predictive performance of method. While I would strongly recommend to indeed add results related to my suggestion, I do not think this should impede publication at any point.

General remarks:

GC1: It is not clear to me how the explained deviance or R2s were calculated for the models. I do not mean the formulas, but instead how the data was separated to calculate them. As there is no explanation given on how it was done, it seems these values refer to the models that included all of the data? From an inference point-of-view, where you wish to understand the relation between the RS parameters and abundance and richness in TreM, this is ok, but this should be supplemented with predictive performance measures, if you want to make statements about prediction (as you do). To estimate predictive performance, deviance and R2’s could, for example, be estimated on a leave-one-plot-out base. This would already provide more insightful values of performance in terms of prediction (even though spatial autocorrelation should actually also be checked for to assess whether neighboring plots hold information about each other, and will hence bias performance in terms of deviance explained or R2s). Anyhow, if you want to know predictive performance, you need to implement a cross-validation schedule in which your model is tested on unseen data.  

Detailed remarks:

- Title: The title is somewhat a mouthful. Consider giving some thought to making it more accessible.

- l.184: shouldn’t be bold.

- Table 1 would probably be better on a landscape page. Everything is rather stretched out because of the Description column being spread over multiple lines. With some luck it fits on 1 landscape page, which would be even better at providing quick insights into what was assessed.

- l.190: I suggest you also use citation 43 here to refer to the R project.

-l.255: As far as I can judge from the text, you cannot currently make statements about predictive performance.

Author Response

Dear reviewer,

Thank you very much for your very positive, relevant and constructive feedback. Please find our replies to your comments in the attached document.

Best regards,

Julian Frey

Reviewer 4 Report

Dear authors,

The reviewer's comments/suggestions are attached as a world file.

The revewer

Author Response

Dear reviewer,

Thank you very much for your extensive and constructive feedback. Please find our replies to your comments in the attached document.

Best regards,

Julian Frey

Round 2

Reviewer 1 Report

This is fine for me.